# Plasma type I collagen α1 chain in relation to coronary artery disease: findings from a prospective population-based cohort and an acute myocardial infarction prospective cohort in Sweden

Filip Hammaréus ![ORCID],[1] Lennart Nilsson ![ORCID],[1] Kwok-Leung Ong,[2] Margareta Kristenson,[1] Karin Festin,[1] Anna K Lundberg,[1] Rosanna W S Chung,[1] Eva Swahn ![ORCID],[1] Joakim Alfredsson,[1] Signe Holm Nielsen,[3,4] Lena Jonasson[1]

SHN and LJ contributed equally.

For numbered affiliations see end of article.

**Correspondence to**
Dr Lennart Nilsson;
lennart.nilsson@liu.se

## ABSTRACT

**Objectives** To investigate the association between type I collagen α1 chain (COL1α1) levels and coronary artery disease (CAD) by using absolute quantification in plasma. Also, to investigate the correlates of COL1α1 to clinical characteristics and circulating markers of collagen metabolism.

**Design** Life conditions, Stress and Health (LSH) study: prospective cohort study, here with a nested case–control design.
Assessing Platelet Activity in Coronary Heart Disease (APACHE) study: prospective cohort study.

**Setting** LSH: primary care setting, southeast Sweden. APACHE: cardiology department, university hospital, southeast Sweden.

**Participants** LSH: 1007 randomly recruited individuals aged 45–69 (50% women). Exclusion criteria was serious disease. After 13 years of follow-up, 86 cases with primary endpoint were identified and sex-matched/age-matched to 184 controls.
APACHE: 125 patients with myocardial infarction (MI), 73 with ST-elevation MI and 52 with non-ST-elevation MI. Exclusion criteria: Intervention study participation, warfarin treatment and short life expectancy.

**Primary and secondary outcome measures** Primary outcome was the association between baseline COL1α1 and first-time major event of CAD, defined as fatal/non-fatal MI or coronary revascularisation after 13 years. Secondary outcomes were the association between the collagen biomarkers PRO-C1 (N-terminal pro-peptide of type I collagen)/C1M (matrix metalloproteinase-mediated degradation of type I collagen) and CAD; temporal change of COL1α1 after acute MI up to 6 months and lastly, correlates between COL1α1 and patient characteristics along with circulating markers of collagen metabolism.

**Results** COL1α1 levels were associated with CAD, both unadjusted (HR=0.69, 95% CI=0.56 to 0.87) and adjusted (HR=0.55, 95% CI=0.41 to 0.75). PRO-C1 was associated with CAD, unadjusted (HR=0.62, 95% CI=0.47 to 0.82) and adjusted (HR=0.61, 95% CI=0.43 to 0.86), while C1M was not. In patients with MI, COL1α1 remained unchanged

## STRENGTHS AND LIMITATIONS OF THIS STUDY

⇒ The study includes two well-characterised cohorts that complement each other in elucidating the role of type I collagen α1 chain (COL1α1) in coronary artery disease (CAD).
⇒ Absolute quantification of COL1α1 in plasma was used to consolidate its novel relationship with incident CAD, as compared with a previous study using a relative quantification method.
⇒ The relationship to other biomarkers reflecting collagen synthesis and degradation, along with markers of inflammation and myocardial injury was investigated to further clarify the role of COL1α1 in CAD.
⇒ Although adjustments for the most common cardiovascular risk factors were performed in our models, there might be residual confounding factors that explain the observed associations.
⇒ The study design and limited sample size (especially women subjects) makes it difficult to draw any definitive mechanistic conclusions from our results along with limiting the generalisability to other populations.

up to 6 months. COL1α1 was correlated to PRO-C1, but not to C1M.

**Conclusions** Plasma COL1α1 was independently and inversely associated with CAD. Furthermore, COL1α1 appeared to reflect collagen synthesis but not degradation. Future studies are needed to confirm whether COL1α1 is a clinically useful biomarker of CAD.

## INTRODUCTION

Cardiovascular disease is the leading cause of death and source of disease burden worldwide. Most clinical manifestations, such as coronary artery disease (CAD), are caused by atherosclerosis.[1] Atherosclerosis is a complex and long-lasting process in which multiple factors such as lipid profile, inflammation

and extracellular matrix (ECM) remodelling contribute to the phenotype of the atherosclerotic plaque. Whether a plaque develops into a vulnerable one, with increased risk of atherothrombotic complications, largely depends on its phenotype.[2] The ECM remodelling process in the arterial wall involves the synthesis and breakdown of ECM proteins such as collagens, proteoglycans and elastin.[3] Excessive or uncontrolled ECM remodelling in CAD has been proposed by previous studies showing that ECM-degrading enzymes, such as matrix metalloproteinases (MMPs), are upregulated in rupture-prone plaques.[4–6] Also, circulating levels of MMPs, particularly MMP-9, have been shown to predict the risk of CAD.[7 8]

Type I collagen is the most abundant protein in the human body and expressed in numerous tissues including skin, bone, tendons and arteries. It consists of two identical polypeptide α1 chains (COL1α1) and one α2 chain resulting in a triple helix molecule. Procollagen is the precursor to mature collagen in tissues and has three major domains: an N-terminal procollagen domain, a central triple helical domain and a C-terminal pro-collagen domain.[9] During the collagen synthesis, the N-terminal and C-terminal pro-collagens are cleaved off from the central domain. These fragments, including the PRO-C1 (N-terminal pro-peptide of type I collagen) biomarker, can reflect the systemic synthesis of type I collagen when measured in plasma.[10] Breakdown fragments of the triple helical domain on the other hand, including the C1M (MMP-mediated degradation of type I collagen) biomarker, can reflect systemic degradation of type I collagen.[11] In atherosclerotic plaques, type I collagen comprises 60% of the total protein content and is the primary component of the fibrous cap.[12 13] The type I collagen content is associated with a stable plaque phenotype. Reversibly, reduced synthesis and/or increased degradation of type I collagen may lead to thinning of the fibrous cap and thereby increase the risk of plaque rupture.[5 14 15] In line with this, circulating markers of collagen type I degradation have been shown to predict incident CAD.[16 17] Whether biomarkers reflecting collagen type I synthesis can be used to predict cardiovascular risk has been studied to a lesser extent. Recently, we performed an exploratory study using relative quantification of 184 cardiovascular disease-related proteins to identify plasma biomarkers that predicted CAD over a 13-year follow-up period in a population-based Swedish cohort.[18] Interestingly, COL1α1 was significantly and inversely associated with the risk of first-time major event of CAD. It also appeared to be the only biomarker which remained significant after multivariate adjustments for common cardiovascular risk factors. This was the first study showcasing the relevance of COL1α1 levels for future CAD risk. However, it is yet to be examined if COL1α1 levels are associated with collagen synthesis/degradation, inflammation and myocardial injury. Also, there are no studies examining COL1α1 changes in the circulation right after an acute CAD event.

Here, the overall aim was to investigate COL1α1 in plasma as a novel biomarker of CAD in a broader context, using two distinct cohorts. In a population-based cohort, we assessed if plasma COL1α1 could be used to determine future risk of CAD. We also assessed whether PRO-C1 and C1M, two biomarkers reflecting type I collagen synthesis and degradation, respectively, were associated with COL1α1 and CAD in the same cohort. In a clinical cohort, we measured plasma COL1α1 in patients with acute myocardial infarction (MI) from admission to 6 months after discharge. In both cohorts, we investigated the correlates of COL1α1, including a number of demographic, clinical and laboratory variables.

## METHODS
### Study population I: the LSH cohort
The longitudinal Life conditions, Stress and Health (LSH) study has been described previously.[8 18] In summary, the LSH cohort consisted of 1007 individuals, aged 45–69 years and evenly distributed by sex, that were randomly recruited from 10 healthcare centres in the County of Östergötland in southeast Sweden. The participation rate was 62.6%. Exclusion criteria were serious physical or psychiatric disease. One participant was excluded due to pancreatic cancer. Data were collected from the end of 2003 to the middle of 2004, using health questionnaires and a visit to the primary healthcare centre for health status, including blood pressure measurements, body mass index (BMI) calculation and collection of fasting venous samples. Venous samples were analysed immediately for routine laboratory analytes including blood lipids. In the questionnaires, participants reported smoking status, medical history and prescription drugs. A composite variable called 'chronic disease' was created including chronic lung disease, cancer and kidney disease. The measurements of C-reactive protein (CRP) (n=250), interleukin (IL)-6 (n=263) and MMP-9 (n=251) in plasma have been described previously.[8] The primary outcome measured during a 13-year follow-up was onset of first-time major CAD event, defined as fatal/non-fatal MI or coronary revascularisation (percutaneous coronary intervention or coronary artery bypass surgery). The outcome data was collected from the Swedish Cause of Death Registry and the Registry of Hospital Admissions managed by the Swedish National Board of Health and Welfare. Six participants did not approve of the linkage of their data to the National Board of Health and Welfare and was therefore excluded from the follow-up. For this study, 86 cases with a primary outcome were identified, 14 on fatal MI, 53 on non-fatal MI, 10 on percutaneous coronary intervention and 9 on coronary artery bypass graft surgery. Mean time to event was 7.0 years (range 0.7–13 years). For a nested case–control design, 184 age-matched and sex-matched controls were identified. Two to three controls of the same sex and born in the same year were randomly selected through computer software to match the cases.

## Study population II: the APACHE cohort

The study of Assessing Platelet Activity in Coronary Heart Disease (APACHE) has been described previously.[19] The APACHE cohort consisted of 125 patients with MI undergoing coronary angiography, of which 73 with ST-elevation MI (STEMI) and 52 with non-ST-elevation MI (NSTEMI), defined according to the global definition of MI.[20] The patients were recruited between January 2009 and August 2011 at the University Hospital Linköping, Sweden. Exclusion criteria were that the patient: already participated in an intervention study, were treated with warfarin before admission, had short life expectancy (less than 6 months) and were unwilling to participate. Information on previous medical history, use of medication and smoking status was collected in a case report form at the time of inclusion. Coronary angiography was performed on all patients with STEMI immediately after admission to hospital and in 98% of patients with NSTEMI within 72 hours. Left ventricular function was assessed by echocardiography 2–3 days after admission. Patients were classified into having normal, mildly reduced, moderately reduced or severely reduced ventricular function according to established American/European guidelines.[21] Blood samples were collected at four time points: at admission (n=122), 3 days (n=117), 7–9 days (n=110) and 6 months after admission (n=106). Laboratory tests such as lipid profile and CRP were analysed according to clinical routine between day 2 and 4 after admission. High-sensitivity troponin T (hsTnT) was assessed 6–8 hours after admission. MMP-9 and IL-6 were analysed in plasma collected from admission (n=102) according to manufacturer's instructions using commercially available ELISA kits (R&D Systems, Minneapolis, Minnesota, USA).

## Quantification of COL1α1 in plasma

Absolute plasma levels of COL1α1 were quantified in both LSH cases (n=86) and controls (n=184), and in the APACHE cohort (n=125). All samples were sodium heparin plasma, except 15 samples in the LSH cohort that were EDTA-plasma (3 cases and 12 controls). Plasma was stored in −70°C from the time of collection until analysis in 2019. After thawing, the sample tubes were centrifuged at 16 000×g for 4 min right before analysis. A Magnetic Luminex Assay (R&D Systems, Catalogue Number LXSAHM) was used to quantify COL1α1 according to manufacturer's instructions. The Magnetic Luminex Assay steps and principles have been described in detail elsewhere.[22] An automatic plate washer, Hydro-Flex (Tecan Trading AG, Switzerland), was used for washing steps. The samples were analysed on a Flexmap 3D analyzer (Luminex, Hertogenbosch, Netherlands) platform using the Bioplex manager software (Bio-Rad) to create a standard curve with a five-parameter logistic curve fit. On every assay plate, with 96 wells per plate, a duplicate of control plasma was analysed to calculate intra-assay and inter-assay coefficients of variation (CV). In the LSH samples, the intra-assay CV ranged from 0.4%

to 7.0% with an inter-assay CV of 14.6%. In the APACHE samples, intra-assay CV ranged from <0.1% to 4.4% and inter-assay CV amounted to 21.1%. Lower limit of quantification (LLOQ) ranged between 735 and 750 pg/mL while upper limit of quantification (ULOQ) ranged between 175 475 and 176 100 pg/mL. Three samples in LSH (two controls and one case) were below the LLOQ and excluded from analysis. The type of anticoagulant did not seem to affect plasma levels of COL1α1, for example, in LSH controls, the levels in sodium heparin plasma (n=172) did not differ significantly from those in EDTA-plasma (n=12), median 4410 pg/mL versus 4490 pg/mL (p=0.21).

Relative measurement of COL1α1 in the LSH cases (n=86) and controls (n=184) was performed using proximity extension assay (PEA) which has been described previously.[18] In short, 184 cardiovascular disease-related proteins, one of which was COL1α1, were measured in plasma. The PEA-technique uses DNA oligonucleotide labelled antibody probes to quantify protein concentrations through real-time PCR technology. The protein concentrations are presented as normalised protein expression (NPX) values. Higher and lower NPX values correspond to higher and lower protein concentrations, respectively, although not being absolute concentration values.

## Quantification of PRO-C1 and C1M in plasma

Two neo-epitope specific biomarkers targeting type I collagen formation (PRO-C1) and type I collagen degradation (C1M) were measured in 255 heparin plasma samples (83 cases and 172 controls) in the LSH cohort using competitive ELISAs developed by Nordic Bioscience (Herlev, Denmark).[10 11] The inter-assay and intra-assay CV were <15% and <10% respectively for both assays. In the case of PRO-C1, 3 samples were below LLOQ while C1M had 21 samples below LLOQ along with 1 sample above ULOQ, we excluded these samples from further analysis.

## Statistical analysis

Descriptive statistics were used to yield means, medians, SDs and IQRs. Medians (IQR) were presented instead of mean (SD) if the variable did not follow distribution of normality in any of the groups presented. Kolmogorov-Smirnov test was used for testing the distribution of normality for the variables. If not stated otherwise, missing values for variables were in single digits and were handled by exclusion from analysis. For comparison between two groups, independent samples t-test was used for normally distributed variables and Mann Whitney U-test for non-normally distributed variables. Correlation analyses were performed using Spearman's analysis. A multiple linear regression model was used when testing for independent predictors of biomarker levels. Variables significantly correlated to biomarker levels in Spearman's analysis were included as non-dependent variables in these models. In APACHE, Friedman analysis was used to explore if there was a significant difference in COL1α1

level across the four time points. To investigate the HR for COL1α1, pro-C1 and C1M in terms of primary outcome (see definition above) in the LSH population, a Cox proportional hazard regression analysis was performed, both with and without covariates. In accordance with our previous study,[18] covariates being adjusted for were established cardiovascular risk factors: sex, age, systolic blood pressure, smoking, body mass index, triglycerides, high-density lipoprotein (HDL) cholesterol, non-HDL cholesterol, CRP, use of anti-hypertensive and lipid-lowering medication and presence of chronic disease. When performing the Cox proportional hazard analysis, we used the $\log_2$ value of COL1α1. Pro-C1 and C1M Cox proportional hazard analyses were performed with $\log_2$ values as well. When assessing the difference in cumulative survival in terms of our primary outcome between groups the log rank test was performed to test for statistical significance along with plotting a Kaplan-Meier curve for visualisation. Two-tailed p<0.05 was used as a cut-off for statistical significance. IBM SPSS, release 28 (IBM Corporation, Armonk, New York, USA) was used for all statistical analyses.

## Patient and public involvement statement

At the time of conducting this research, patients and/or the public were not involved in the design, conduct, reporting or dissemination of the research.

## RESULTS

### Plasma levels of COL1α1 and their association with incident CAD in the LSH cohort

Demographic, clinical and laboratory characteristics of the LSH cases (n=86) and controls (n=184) are presented in table 1. The mean age was 61 years and around 64% were men. The cases smoked more, had higher blood pressures, higher levels of non-HDL cholesterol, triglycerides, CRP and MMP-9 and lower levels of HDL cholesterol compared with controls. As shown in table 2, the absolute concentrations of COL1α1 measured by Luminex were significantly lower in LSH cases compared with controls. Furthermore, the absolute concentrations of COL1α1 exhibited a significant correlation with the PEA-based relative concentrations of COL1α1 (r=0.69, p<0.001, see online supplemental figure 1). The PEA-based data were previously published by Ong et al.[18]

We next proceeded to assess whether absolute concentrations of COL1α1 were associated with incident CAD at 13-year follow-up (table 2). When performing Cox proportional hazard analysis, COL1α1 levels showed significant association with incident CAD in the unadjusted model (HR=0.69, 95% CI=0.56 to 0.87, p=0.001). In the adjusted model, COL1α1 levels remained significantly associated with incident CAD (HR=0.55, 95% CI=0.41 to 0.75, p<0.001). Furthermore, when the LSH participants were separated into two groups based on their COL1α1 values, above or below the median level (≥ or < 4010 pg/mL), their cumulative survival in terms of CAD outcome differed significantly (p=0.002, figure 1).

**Table 1** Baseline characteristics of cases and controls in the LSH cohort

| Characteristics | Controls (n=184) | Cases (n=86) | P value |
|---|---|---|---|
| Age, years | 60.6 (6.4) | 60.6 (6.6) | 0.84 |
| Women, n (%) | 66 (36) | 31 (36) | >0.9 |
| Smokers, n (%) | 19 (11) | 24 (30) | <0.001 |
| BMI, kg/m² | 26.8 (3.6) | 27.8 (3.8) | 0.048 |
| Systolic blood pressure, mm Hg | 135 (18) | 146 (24) | <0.001 |
| Antihypertensive medication, n (%) | 32 (17) | 30 (35) | 0.002 |
| Lipid-lowering medication, n (%) | 9 (4.9) | 11 (13) | 0.027 |
| Diabetes, n (%) | 14 (7.6) | 8 (9.3) | 0.64 |
| Angina pectoris, n (%) | 4 (2.2) | 12 (14) | <0.001 |
| Serious chronic illness, n (%) | 13 (7.1) | 9 (11) | 0.34 |
| HDL-cholesterol, mmol/L* | 1.50 (0.50) | 1.40 (0.42) | 0.001 |
| LDL-cholesterol, mmol/L | 3.54 (0.82) | 3.64 (0.90) | 0.39 |
| Non-HDL cholesterol, mmol/L | 4.11 (0.89) | 4.36 (0.91) | 0.036 |
| Triglycerides, mmol/L* | 1.10 (0.66) | 1.40 (0.92) | <0.001 |
| CRP, mg/L* | 0.835 (1.7) | 1.56 (2.4) | 0.01 |
| IL-6, pg/mL* | 0.560 (2.25) | 1.68 (2.5) | 0.097 |
| MMP-9, ng/mL* | 30.7 (32) | 39.5 (31) | 0.036 |

Data expressed as mean (SD) or median (IQR).
*if data do not possess distribution of normality. P values from comparing controls and cases in terms of characteristics.
BMI, body mass index; CRP, C-reactive protein; HDL, high density lipoprotein; IL-6, interleukin-6; LDL, low density lipoprotein; LSH, Life conditions, Stress and Health; MMP-9, matrix metalloproteinase-9.

### Plasma levels of PRO-C1 and C1M and their associations with incident CAD in the LSH cohort

As further shown in table 2, PRO-C1 levels were significantly lower in cases compared with controls in LSH, while C1M levels did not differ significantly between the groups. PRO-C1 levels showed a significant association with CAD outcome both in the unadjusted (HR=0.62, 95% CI=0.47 to 0.82, p=0.001) and adjusted model (HR=0.61, 95% CI=0.43 to 0.86, p=0.005). On the other hand, C1M levels did not show any significant association with CAD outcome. Cumulative survival differed significantly in terms of CAD outcome when the LSH cohort was split based on PRO-C1 levels into groups with above or below median PRO-C1 levels (≥ or < 71.9 ng/mL) (p<0.001, figure 1).

**Table 2** Circulating levels of COL1α1, PRO-C1, C1M and their association with incident coronary artery disease at 13-year follow-up in the LSH cohort

| Biomarker | Median (IQR) level | | | | Unadjusted model | | Adjusted model* | |
|---|---|---|---|---|---|---|---|---|
| | Control (n=184†) | Case (n=86†) | Difference | P value | HR (95% CI) | P value | HR (95% CI) | P value |
| COL1α1 (pg/mL) | 4430 (2900) | 3810 (3400) | 16% | 0.002 | 0.69 (0.56 to 0.87) | 0.001 | 0.55 (0.41 to 0.75) | <0.001 |
| PRO-C1 (ng/mL) | 77.6 (43) | 59.7 (39) | 30% | <0.001 | 0.62 (0.47 to 0.82) | 0.001 | 0.61 (0.43 to 0.86) | 0.005 |
| C1M (ng/mL) | 22.0 (16) | 23.9 (19) | 8.6% | ns | 1.57 (0.57 to 4.34) | ns | 0.92 (0.29 to 3.37) | ns |

Association expressed as HRs per one unit increase in log2-tranformed unit from Cox proportional hazard analysis.
*Covariates being adjusted for were sex, age, systolic blood pressure, smoking, body mass index, triglycerides, high-density lipoprotein (HDL) cholesterol, non-HDL cholesterol, C-reactive protein (CRP), use of anti-hypertensive and lipid-lowering medication and presence of chronic disease.
†For pro-C1 and C1M, n=172 and n=83 for controls and cases, respectively.
C1M, MMP-mediated degradation of type I collagen; COL1α1, collagen type I α1 chain; HR, hazard ratio; LSH, Life conditions, Stress and Health; ns, non-significant; PRO-C1, N-terminal pro-peptide of type I collagen.

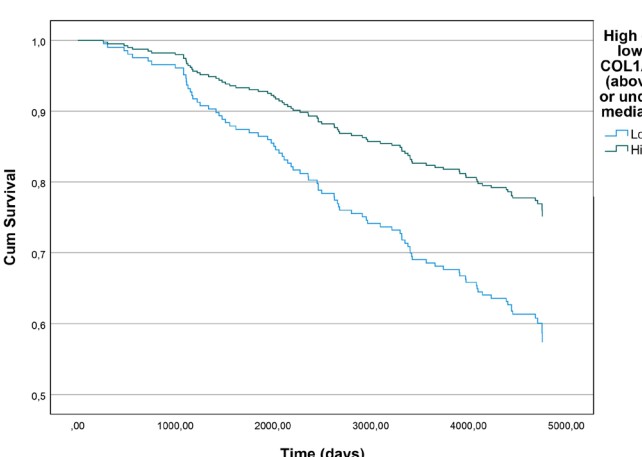

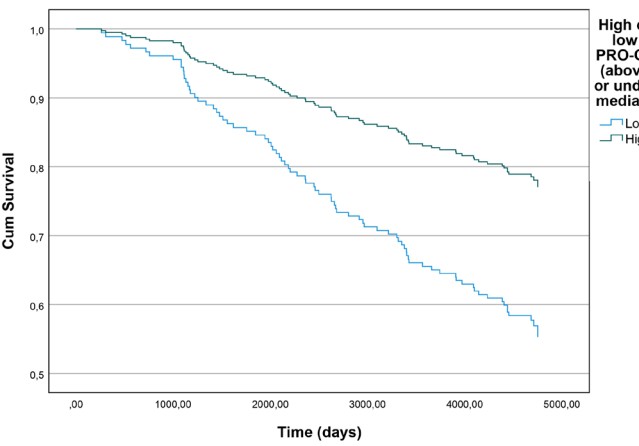

**Figure 1** Kaplan-Meier survival curves showing cumulative survival in terms of primary outcome (MI or invasive coronary intervention) between participants with high versus low (above or under median) COL1α1 or PRO-C1 values in the LSH cohort. COL1α1, collagen type I α1 chain; Cum, cumulative; LSH, Life conditions, Stress and Health; MI, myocardial infarction; PRO-C1, N-terminal pro-peptide of type I collagen.

### Temporal changes of plasma COL1α1 in the APACHE cohort

Demographic, clinical and laboratory characteristics of the APACHE cohort, 73 patients with STEMI and 52 patients with NSTEMI, are shown in table 3. The mean age was 66 years and around 70% were men. The COL1α1 levels did not change significantly from admission to 6 months in patients with neither STEMI nor NSTEMI. In terms of different time points, COL1α1 median (IQR) value in the APACHE cohort was 3590 (3200), 3780 (3100), 3570 (3300) and 3330 (2800) pg/mL at arrival, after 3 days, after 7–9 days and after 6 months, respectively (p=0.92). As shown in figure 2, COL1α1 levels in the APACHE cohort (at admission) were comparable to COL1α1 levels in the LSH cases.

### Correlates of COL1α1, PRO-C1 and C1M

Among baseline characteristics in the LSH cohort, COL1α1 showed correlations with female sex, HDL cholesterol, diabetes and angina pectoris (online supplemental table 1). In a multiple regression model, only female sex remained significantly associated with COL1α1 levels (*beta*=0.21, p=0.001). As further shown in online supplemental table 1, PRO-C1 correlated with female sex, BMI, angina pectoris, triglycerides, HDL cholesterol, IL-6 and MMP-9. In a multiple regression model, only female sex remained significantly associated with PRO-C1 levels (*beta*=0.23, p=0.001). Further, C1M correlated to female sex, age, BMI, triglycerides, non-HDL cholesterol, CRP, diabetes and IL-6. However, in a multiple regression model, only CRP (*beta*=1.68, p<0.001) and non-HDL cholesterol (*beta*=1.80, p=0.032) remained significantly related to C1M levels. When examining the intercorrelations between COL1α1, PRO-C1 and C1M in the LSH cohort, PRO-C1 and COL1α1 showed a strong correlation (*r*=0.73, p<0.001) while C1M did not correlate to either COL1α1 or PRO-C1.

In the APACHE cohort, female sex showed a significant but weak correlation with COL1α1 concentrations

**Table 3** Baseline characteristics of 125 patients with acute MI in the APACHE cohort

| Characteristics | Total (n=125) | STEMI (n=73) | NSTEMI (n=52) | P value |
|---|---|---|---|---|
| Age, years | 66.0 (12) | 65.9 (13) | 66.1 (11) | 0.86 |
| Women, n (%) | 37 (30) | 25 (34) | 12 (23) | 0.18 |
| Smokers, n (%) | 34 (27) | 25 (34) | 9 (17) | 0.037 |
| BMI, kg/m$^2$* | 26.3 (4.4) | 26.2 (4.2) | 26.3 (4.1) | 0.59 |
| Systolic blood pressure, mm Hg | 148 (28) | 141 (26) | 159 (28) | 0.001 |
| Diastolic blood pressure, mm Hg | 83 (17) | 81 (15) | 86 (20) | 0.14 |
| Lipid-lowering medication, n (%) † | 31 (25) | 13 (18) | 18 (35) | 0.033 |
| Diabetes, n (%) | 16 (13) | 7 (9.5) | 9 (17) | 0.18 |
| LV-function, normal/mildly reduced, n (%) | 109 (87) | 60 (82) | 49 (94) | <0.001 |
| hsTnT (at 6–8 hours), ng/mL * | 9.12 (91) | 7.90 (130) | 17.5 (71) | 0.68 |
| HDL-cholesterol, mmol/L* | 1.10 (0.37) | 1.10 (0.39) | 1.10 (0.45) | 0.59 |
| LDL-cholesterol, mmol/L | 2.84 (0.90) | 2.94 (0.89) | 2.67 (0.89) | 0.12 |
| Non-HDL cholesterol, mmol/L * | 3.49 (1.3) | 3.50 (1.3) | 3.40 (1.5) | 0.51 |
| Triglycerides, mmol/L * | 1.45 (0.87) | 1.40 (0.75) | 1.50 (1.1) | 0.50 |
| CRP, mg/L* | 3.66 (6.0) | 4.44 (5.3) | 3.16 (7.0) | 0.34 |
| IL-6, pg/mL * | 3.58 (5.0) | 4.36 (5.8) | 2.64 (3.0) | 0.005 |
| MMP-9, ng/mL* | 263 (210) | 253 (170) | 310 (230) | 0.62 |
| COL1α1, pg/mL* | 3590 (3200) | 3670 (3900) | 3580 (2400) | 0.31 |

Statistics expressed as mean (SD) or median (IQR).
*if data do not possess distribution of normality. P values from comparing patients with STEMI and NSTEMI in terms of characteristics. Laboratory variables represent levels at admission (if not otherwise indicated).
†Defined as statin use at arrival to cardiology department.
APACHE, Assessing Platelet Activity in Coronary Heart Disease; BMI, body mass index; COL1α1, collagen type I α1 chain; CRP, C-reactive protein; HDL, high density lipoprotein; hsTnT, high-sensitivity Troponin T; IL-6, interleukin-6; LDL, low density lipoprotein; LV, left ventricular; MI, myocardial infarction; MMP-9, matrix metalloproteinase-9; NSTEMI, non-ST-elevation MI; STEMI, ST-elevation MI.

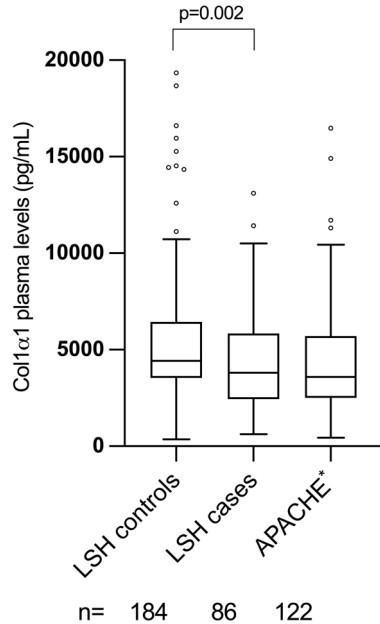

**Figure 2** Box plots showing COL1α1 plasma levels (pg/mL) measured by the Luminex Assay among participants in LSH and APACHE. P value based on Mann-Whitney U test. *At admission. APACHE, Assessing Platelet Activity in Coronary Heart Disease; COL1α1, collagen type I α1 chain; LSH, Life conditions, Stress and Health study.

in bivariate correlation analyses at 6 months (r=0.19, p=0.047). COL1α1 did not showcase any significant correlations to IL-6, MMP-9 or CRP in LSH nor in APACHE. Neither was it correlated to hsTnT or to left ventricular dysfunction in APACHE.

As shown in online supplemental figure 2, women had significantly higher COL1α1 levels than men in the LSH cohort. Also, female controls had significantly higher COL1α1 levels than female cases. Furthermore, PRO-C1 levels were higher in women than in men, 85.6 (60) versus 67.3 (32) ng/mL (p=0.001) while C1M levels did not differ between the sexes (data not shown). In the APACHE cohort, the difference in COL1α1 levels at admission between men and women did not reach significance but was significantly different at 6 months, median (IQR) 3200 (2600) versus 3840 (3700) pg/mL, respectively (p=0.048).

## DISCUSSION

The present study showed that COL1α1 in plasma was independently and inversely associated with future risk of CAD. PRO-C1, a marker for type I collagen synthesis, showed a similar relationship to CAD outcome while C1M, a marker for type I collagen degradation, did not.

Further investigation of COL1α1 in an MI population showed that levels remained stable up to 6 months after the acute event without any relationship to inflammation or myocardial injury.

In the population-based cohort, COL1α1 levels in plasma were significantly lower in those who developed CAD compared with those who did not. A cox proportional hazard analysis confirmed the association between lower COL1α1 levels and CAD outcome, even after multiple adjustment. The results confirm, and extend, the previous findings by Ong et al.[18] However, in that explorative study, the measurements of COL1α1 were based on relative quantification, and it was therefore important to validate the results by a quantifying assay. Prior to the study by Ong et al,[18] there were no studies available on COL1α1 in a CAD setting, except for one study reporting that a COL1α1 gene polymorphism occurred more frequently in MI survivors, as compared with age-matched controls.[23]

Similar to COL1α1, PRO-C1, a biomarker of type I collagen synthesis, was inversely associated with CAD in both unadjusted and adjusted models while C1M, a biomarker for type I collagen breakdown, did not show any association. In line with this, COL1α1 showed a strong correlation with PRO-C1 indicating that both are markers of type I collagen synthesis. COL1α1 is the α1 chain partly making up the type I collagen molecule, while PRO-C1 is the internal epitope in the N-terminal pro-peptide of type I collagen. Hence, they can be regarded as products from two different stages of type I collagen synthesis.[9 10] To our knowledge, no previous studies have investigated PRO-C1 in the context of CAD. On the other hand, C1M has been associated with increased risk of cardiovascular events. One previous study reported that elevated C1M levels predicted MI in a prospective cohort of postmenopausal women (n=5450, median age 71 years) with a median follow-up time of 14 years.[17] Another study with a follow-up of 6 years reported that elevated C1M levels predicted cardiovascular events and mortality in patients who had undergone carotid endarterectomy (n=473, median age 72 years, 64% men).[16] The discrepancy in results between these previous studies and the present study as regards C1M could be due to several reasons. First, the two previous studies measured C1M in serum while we used plasma, a distinction that can lead to different outcomes with examples from other biomarkers such as MMPs.[24] Moreover, the study populations are not comparable in terms of size and composition. For example, the study by Holm Nielsen et al[16] included patients who had undergone carotid endarterectomy yielding an older cohort with high atherosclerotic burden compared with our population-based cohort.

In the present study, we also investigated the temporal pattern of COL1α1 levels in patients with MI by performing serial measurements from admission up to 6 months after the acute event. COL1α1 did not show any temporal changes over time in patients with STEMI or NSTEMI, neither did it show any correlation with inflammatory markers or hsTnT in the early phase. Obviously, COL1α1 levels in plasma were not associated with acute atherothrombosis, inflammation or myocardial injury. Interestingly, COL1α1 values in patients with acute MI in the APACHE cohort were found to be comparable to COL1α1 values in individuals who subsequently developed incident CAD in LSH, as shown in figure 2. However, this comparison should be made with caution since it is based on two distinct cohorts.

Women had significantly higher levels of COL1α1 and PRO-C1 compared with men in the LSH cohort. Female sex was also the only determinant that remained independently associated with COL1α1 and PRO-C1 in a multiple regression model including baseline variables. Collagen homeostasis has been shown to be affected by oestrogen.[25] PRO-C1 and other bone turnover markers are known to increase in women after menopause while there is little increase in men with age.[26] This may partly explain why women in our population-based cohort had higher levels of COL1α1 and PRO-C1 than men. The sex difference in COL1α1 was also observed in the APACHE cohort, however these data should be interpreted with caution due to the limited number of female subjects in APACHE.

The major limitation of our study is the limited sample size, especially as regards women. Also, our cohorts reflected the Swedish population, which may limit the generalisability of results to other populations. Furthermore, in the LSH-cohort, case–control matching was performed based only on age and sex, which might result in differences between the groups that could affect the study outcome. To address this issue, other factors known to be of major importance for CAD risk were included in adjustment models. Still, it is possible that we have residual confounding factors contributing to the association between COL1α1 and risk of CAD. A recent study showed that COL1α1 was upregulated in left ventricular heart tissue from patients with heart failure and that plasma levels were increased in those who were candidates for heart transplant.[27] However, the MI cohort in our study included only 16 patients with moderate or severe left ventricular dysfunction, which does not permit us to draw conclusions about the relationship between COL1α1 and heart failure.

Based on our results, it can be speculated that plasma levels of COL1α1 and PRO-C1 reflect the ability to form type I collagen in arterial tissue, thereby influencing the development and fate of atherosclerotic plaques and risk of MI. However, bones and tendons are other sources of COL1α1in the circulation. Also, the liver may leak ECM components to the bloodstream under pathological conditions.[28] Whether COL1α1 itself has a causal role in the development of CAD is unclear. Interestingly, a few previous clinical studies have investigated the effect of collagen supplement therapies on atherosclerosis-related factors. These studies, although small and with short intervention periods, have shown positive effects on LDL (low density lipoprotein) cholesterol/HDL cholesterol

 

ratio, toxic advanced glycation end-products, cardio-ankle vascular index and brachial-ankle pulse wave velocity.[29 30]

To summarise, we confirm that absolute concentrations of COL1α1 in plasma are independently and inversely associated with future risk of CAD, but does not seem to change in an acute coronary event setting. COL1α1 in plasma appears to reflect collagen synthesis, but not collagen degradation, inflammation or myocardial injury. Larger studies are needed to confirm the clinical usefulness of COL1α1 as a marker of coronary atherosclerosis and/or predictor of incident CAD.

**Author affiliations**
¹Department of Health Medicine and Caring Sciences, Linkoping University, Linkoping, Sweden
²Faculty of Medicine and Health, NHMRC Clinical Trials Centre, The University of Sydney, Sydney, New South Wales, Australia
³Department of Biotechnology and Biomedicine, Technical University of Denmark, Lyngby, Denmark
⁴Nordic Bioscience, Herlev, Denmark

**Acknowledgements** We would like to acknowledge Bettina Jung and Sofie Madsen for their technical support. The research was supported by a grant from the County Council of Östergötland, Sweden, aimed towards scientists early in their career; Academy of Health Care, County Council of Jönköping (Futurum); and The Danish Research Foundation and Innovation Fund Denmark.

**Contributors** FH wrote the manuscript. FH, AKL, LJ, MK, K-LO, ES and JA participated in study design. FH, RWSC and AKL prepared plasma samples and performed the Luminex-assay on COL1α1. SHN measured C1M and PRO-C1 levels in the LSH cohort. FH, KF, LN and JA participated in data analysis. All authors participated in data interpretation and critical revision of the manuscript. LJ is responsible for the overall content as the guarantor.

**Funding** The LSH study was funded by the Swedish Research Council (2004–1881) and the Swedish Heart and Lung Foundation (2004053). The APACHE study received financial support from ALF Region Östergotland (LIO 131 471). The research was supported by the County Council of Östergötland, specifically with a grant aimed towards scientists early in their career (RÖ-910951); Academy of Health Care, County Council of Jönköping (N/A); and The Danish Research Foundation (N/A) and Innovation Fund Denmark (N/A). Further, we acknowledge the support from Linköping University.

**Competing interests** SHN is a full-time employee at Nordic Bioscience A/S. Nordic Bioscience is a privately-owned, small–medium size enterprise (SME) partly focused on the development of biomarkers including PRO-C1 and C1M. No fees, bonuses or other benefits were rewarded for the work described in the manuscript. SHN holds stocks in Nordic Bioscience A/S.

**Patient and public involvement** Patients and/or the public were not involved in the design, or conduct, or reporting, or dissemination plans of this research.

**Patient consent for publication** Not applicable.

**Ethics approval** The LSH and APACHE studies have been approved by The Regional Ethical Review Board in Linköping (Dnr 02–0324 and M45-08 respectively). All study participants have given written informed consent.

**Provenance and peer review** Not commissioned; externally peer reviewed.

**Data availability statement** Data are available upon reasonable request.

**ORCID iDs**
Filip Hammaréus http://orcid.org/0000-0003-4953-6124
Lennart Nilsson http://orcid.org/0000-0001-7887-7978
Eva Swahn http://orcid.org/0000-0002-2608-2062

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
