## [Reviewer comments · BMJ Open]

ARTICLE DETAILS

TITLE (PROVISIONAL)	Plasma type I collagen α 1 chain in relation to coronary artery disease: findings from a prospective population-based cohort and an acute myocardial infarction prospective cohort in Sweden
AUTHORS	Hammaréus, Filip; Nilsson, Lennart; Ong, Kwok-Leung; Kristenson, Margareta; Festin, Karin; Lundberg, Anna; Chung, Rosanna; Swahn, Eva; Alfredsson, Joakim; Holm Nielsen, Signe; Jonasson, Lena

VERSION 1 – REVIEW

REVIEWER	Narayan, Pradeep Bristol Heart Institute
REVIEW RETURNED	16-Apr-2023

GENERAL COMMENTS	This is an interesting study, however, there are some points that need further. 1. When was the sample for Plasma levels of COL1α1 taken from the participants and the controls?2. The authors mention that they chose age and gender matched controls. Why did the authors not carry out a propensity matching that could have eliminated other differences observed in the 2 groups in Table 1? This has led to greater number of smokers in the case group which is an important risk factor for coronary artery disease and events.3. What was the rationale of including APACHE cohort? This appears in the methodology for the first time. The authors have to describe why correlation with this group is important and should add this in the introduction.4. What was the primary outcomes in the study? The authors provide different definitions in different section.5. In the abstract they write that the primary outcome was the association between baseline COL1α1 and CAD after 13 years (LSH).6. In methods, the authors mention that the “primary outcome measure during 13 years of follow-up was onset of first-time major event of CAD, defined as fatal/non-fatal MI or coronary revascularization (percutaneous coronary intervention or coronary artery bypass surgery”.7. In statistical analysis they write- “When assessing the difference in cumulative survival in terms of our primary outcome between groups the log rank test was performed”8. Also, could they breakdown the event of fatal/non-fatal MI or coronary revascularization (percutaneous coronary intervention or coronary artery bypass surgery among the cases.
--

	9. Language needs some improvement too. This is an interesting study, but it almost feels that it is a combination of 2 separate studies- the LSH and the APACHE. The study (APACHE) has not even mentioned in the discussion. The strength lies in the long follow-up but the study findings need to be presented more lucidly.
--	--

REVIEWER	Li, Yubo Tianjin University of Traditional Chinese Medicine
REVIEW RETURNED	24-Apr-2023

GENERAL COMMENTS	There is no doubt that research on cardiovascular disease is very interesting, and I mainly have the following suggestions for this research  1. There are few pictures in this manuscript, and did the author consider increasing the number of images so that the experimental results can be displayed more visually? 2. The control group had more than twice as many samples as the LSH group, and the comparison seemed unbalanced 3. If there are more smokers in male cases, does smoking also affect the outcome? If not, then by what means did the author confirm it 4. The structure of the articles in this manuscript does not seem to be very standardized, and the author is advised to reorganize the manuscript
--

REVIEWER	Senn, Stephen The University of Sheffield, School of Health and Related Research
REVIEW RETURNED	27-May-2023

GENERAL COMMENTS	 1) You might like to consult the EQUATOR case-control guidelines https://www.goodreports.org/reporting-checklists/strobe-case-control/ and check that your reporting conforms with this.
 2) I am not entirely convinced that matching by age was appropriate. You might like to consult Mansournia et al.
 3) Regardless as to whether or not it was appropriate, please give more details on how it was done.
  a) How close did ages have to be to constitute a match?
 b) The average number of controls per case was 2.14 but what was the distribution? A little table would be useful. Reference
 MANSOURNIA, M. A., JEWELL, N. P. & GREENLAND, S. 2018. Case-control matching: effects, misconceptions, and recommendations. Eur J Epidemiol, 33, 5-14.
---

VERSION 1 – AUTHOR RESPONSE

Response to Reviewer: 1, Dr. Pradeep Narayan, Bristol Heart Institute

1. *When was the sample for Plasma levels of COL1α1 taken from the participants and the controls?*

Response: The venous samples of the cases and controls in LSH were collected during the years 2003-2004 and for APACHE 2009-2011. Plasma was immediately frozen down in -70 degrees Celsius until analysis of COL1a1 levels in the summer of 2019. This has now been added and clarified in the methods section of the manuscript.

2. *The authors mention that they chose age and gender matched controls. Why did the authors not carry out a propensity matching that could have eliminated other differences observed in the 2 groups in Table 1? This has led to greater number of smokers in the case group which is an important risk factor for coronary artery disease and events.*

Response: Now in hindsight, we agree that a propensity matching would have been a legitimate and a useful tool to eliminate differences between groups. Carrying out this particular study, the matching of controls had already been performed by our co-authors responsible for the LSH cohort. Age and sex were at this time considered the most important factors that would contribute to the main outcome. As you mention, smoking is also an important CVD risk factor among other characteristics such as lipid profile. We addressed this fact through adjustments in our main analysis, as described in Methods. This point has now been added to the study limitations section in discussion. Lastly, an important aim with this study was to confirm previous findings presented by Ong et al., motivating a similar study design for comparability between studies. Please also see our response regarding smoking below to reviewer #3. We would also like to point out the pit-falls in propensity matching presented by Mansournia et al. (reference below): *“Propensity-score methods are sometimes promoted to address the concerns we have discussed. Even in cohort studies, however, propensity-score matching may lead to overadjustment and variance inflation, or poor control of strong confounders [60,61,62], and can also generate spurious results in case-control studies [63]. Thus propensity scoring does not address the need to consider causal structure, associational strength, and potential artefacts when formulating a matching protocol.”*

3. *What was the rationale of including APACHE cohort? This appears in the methodology for the first time. The authors have to describe why correlation with this group is important and should add this in the introduction.*

Response: For us, the APACHE cohort adds value in the manuscript for two main reasons. Quoting the main aim of the study, “the overall aim was to investigate COL1a1 in plasma as a novel biomarker of CAD in a broader context using two distinct cohorts” which stems from results previously mentioned in the introduction where COL1a1 for the first time was found to be associated with future CAD. Because no previous research has been performed on COL1a1 in a CAD setting, we wanted to explore this biomarker both in a “pre-event” setting (long-term) and an “acute event” setting (short-term). APACHE gave us the opportunity to examine the latter and based on the results, we conclude that COL1a1 is suited more as a pre-event marker rather than a marker that changes during an acute event. The cohort is implicitly mentioned on page 4-5 line 150-153 referring to aim 3. In the same way, LSH is not mentioned by name in the introduction either. Another way APACHE adds value is by offering more participants where correlates between COL1a1 and other clinical variables can be tested. We agree that the rationale for including APACHE could be further elucidated in the introduction, this has now been added to page 4 line 142-143.

4. *What was the primary outcomes in the study? The authors provide different definitions in different section.*

Response: We agree that there is a lack of consistency in describing the primary outcome. Changes have now been made in the abstract to keep the definition consistent throughout the manuscript.

5. *In the abstract they write that the primary outcome was the association between baseline COL1a1 and CAD after 13 years (LSH).*

Response: Please, see above.

6. *In methods, the authors mention that the “primary outcome measure during 13 years of follow-up was onset of first-time major event of CAD, defined as fatal/non-fatal MI or coronary revascularization (percutaneous coronary intervention or coronary artery bypass surgery”.*

Response: Please, see above.

7. *In statistical analysis they write- "When assessing the difference in cumulative survival in terms of our primary outcome between groups the log rank test was performed.*

Response: Please, see above.

8. *Also, could they breakdown the event of fatal/non-fatal MI or coronary revascularization (percutaneous coronary intervention or coronary artery bypass surgery among the cases.)*

Response: This is an important comment and we believe the reader gets a better understanding of the cases if they know how the proportions look like in this group. This has now been added to the end of the LSH section under methods. To recapitulate, 14 were included on fatal MI, 53 on non-fatal MI, 10 on PCI (no MI) and 9 on CABG (no MI). The composite outcome was determined by the PI of LSH before this study was conducted and it was motivated by an *a priori* power calculation for the main outcome measure in LSH. However, we believe that the event of fatal/non-fatal MI or coronary revascularization generally captures significant CAD in a fulfilling way and it improves the statistical power in this setting as well.

9. *Language needs some improvement too.*

Response: The article has been proof-read by all co-authors along with other researchers well familiarized with the English language, most do however not practice English as first language which could be apparent. Thus, we have taken the article through another round of language changes which can be found in the updated text file.

This is an interesting study, but it almost feels that it is a combination of 2 separate studies- the LSH and the APACHE. The study (APACHE) has not even mentioned in the discussion. The strength lies in the long follow-up but the study findings need to be presented more lucidly.

Response: Yes, in this study we combine two cohorts in order to better understand the role of plasma COL1 α 1 in CAD. We kindly refer to our response on point 3 above for rationale of using the APACHE cohort as well. We have made efforts to further clarify the use of two distinct cohorts in the text. Regarding the discussion, findings from the APACHE cohort are now clarified.

Response to Reviewer: 2, Prof. Yubo Li, Tianjin University of Traditional Chinese Medicine

1. *There is no doubt that research on cardiovascular disease is very interesting, and I mainly have the following suggestions for this research 1. There are few pictures in this manuscript, and did the author consider increasing the number of images so that the experimental results can be displayed more visually?*

Response: This is a good point and we have thought about how to best visualize the results. For every major result of the study, we present them in a figure or table, which we believe illustrate the findings appropriately.

2. *The control group had more than twice as many samples as the LSH group, and the comparison seemed unbalanced*

3. *If there are more smokers in male cases, does smoking also affect the outcome? If not, then by what means did the author confirm it*

Response: Yes, this is correct. We chose twice as many controls to increase statistical power in our analysis which is a method widely used for case-control studies, please see Grimes et al. below. Although not addressing biases, increasing the amounts of controls per case when the number of cases are limited is often a cost-effective way to detect differences of importance.

Smoking does affect outcome. First, looking at table 1, page 18, one can notice that 30% of individuals developing first time CVD smoked compared to 11% in the control group ($p < 0.001$). Secondly, smoking is a well-established CVD risk factor. Thirdly, when performing a cox regression analysis with smoking status as independent variable and first-time CVD (the primary outcome of the study) as dependent, HR = 2.7 ($p < 0.001$) for smokers. Stemming

from this, we chose to adjust for smoking in our main analysis, decreasing the effect of this confounder on the main results.

4. *The structure of the articles in this manuscript does not seem to be very standardized, and the author is advised to reorganize the manuscript*

Response: The article is structured according to BMJ open's standard. Furthermore, we try to use subheadings in order to provide the reader with even more structure. In the discussion part, although no subheadings are used, we try to organize it in the same order as the results are presented along with paragraphs presenting main findings, limitations etc. We kindly also refer to the STROBE checklist which we attach upon the editor's request. Here, we point out where in the article each item of the STROBE checklist can be found.

Response to Reviewer: 3, Prof. Stephen Senn, The University of Sheffield

- 1) You might like to consult the EQUATOR case-control guidelines <https://www.goodreports.org/reporting-checklists/strobe-case-control/> and check that your reporting conforms with this.

Response: Thank you for referring to the EQUATOR guidelines which we think can be very useful for reporting all necessary information in a case-control study. Please see our attached STROBE checklist which is attached upon the request of the editor where similar information is provided. Upon this cross-checking with STROBE checklist items, we got a chance to clarify and extend certain aspects in the article.

- 2) *I am not entirely convinced that matching by age was appropriate. You might like to consult Mansournia et al.*

Response: We think Mansournia et al. put forward important difficulties in case-control matching and we read the article with great interest. In the article, the authors states "*In conclusion, we concur with advice that matching should be used with great caution, especially in case-control studies [1]. Variables expected to be strong confounders (like age and sex) are good candidates for direct matching, whereas weak confounders may be better addressed via subsequent model-based adjustments, while matching or adjustment for variables unrelated to disease is best avoided*". They later conclude "*The most practical option may often be to match only on age and sex...*". In our case, both sex and age are strongly associated to study outcome which is why this approach was chosen when the LSH cohort was assembled long before this study was conducted. However, without elaborating this further, Mansournia et al. mention that matching for covariates (association to outcome or exposure) rather than confounders (association to outcome and exposure), could introduce bias. In our case, age has this property, linking strongly to outcome but not strongly to exposure (COL1a1 levels). For future studies, there could therefore be an important discussion if age-matching is necessary if examining COL1a1 solely.

- 3) *Regardless as to whether or not it was appropriate, please give more details on how it was done.*

a) *How close did ages have to be to constitute a match?*

b) *The average number of controls per case was 2.14 but what was the distribution? A little table would be useful.*

Reference

Response: We aimed for twice as many controls as cases to improve statistical power, an approach commonly used in case-control studies, please see Grimes et al. below. Two controls were randomly generated per each case through computer software. The age-matched controls were born in the same year as their respective case. We ended up including some extra controls, also randomly generated, given we had empty plate spots in our protein quantification analysis. As a result, most cases had two matched controls and a few had

three. This distribution has now been clarified in the methods section.

References:

Grimes DA, Schulz KF. Compared to what? Finding controls for case-control studies. Lancet. 2005 Apr 16-22;365(9468):1429-33.

Mansournia MA, Jewell NP, Greenland S. Case-control matching: effects, misconceptions, and recommendations. Eur J Epidemiol. 2018 Jan;33(1):5-14.

VERSION 2 – REVIEW

REVIEWER	Narayan, Pradeep Bristol Heart Institute
REVIEW RETURNED	08-Aug-2023

GENERAL COMMENTS	Overall, the authors have addressed the concerns raised and provided reasonable explanations. They also reference relevant literature to support their decisions and acknowledged the limitations. There are few minor points 1. Lines 336-338- the authors mention that – In the APACHE cohort, female sex was the only variable that showed a significant correlation with COL1A1 concentrations in bivariate correlation analyses, though only at 6 months ($r = 0.19$, $p = 0.047$). To be honest, the p-value if limited to 2 decimal points would become 0.05 and a $r = 0.19$ shows a very weak correlation. So, it might be best to rephrase the statement as it does not really show a “significant correlation”. Also, even though it is a very minor issue the p-value in the figure 2 is 0.048 but in the text it is 0.047. 2. The authors have defended the choice of the APACHE cohort by stating that they wanted to study this (COL1A1 biomarker both in a “pre-event” setting (LSH) and an “acute event” setting (APACHE). In the conclusion, however, they only clearly report the findings of the LSH cohort. They should ideally summarize that COL1A1 is suited more as a pre-event marker rather than a marker that changes during an acute event. 3. Just out of interest it appears that all the biomarkers have been tabulated (table2) as median and IQR. Is it because the data was not normally distributed? Besides, it might be more informative to provide the data in the form Median (Quartile 1: quartile 3).
---

REVIEWER	Li, Yubo Tianjin University of Traditional Chinese Medicine
REVIEW RETURNED	14-Aug-2023

GENERAL COMMENTS	After revisions, I think this manuscript is ready for publication.
--

VERSION 2 – AUTHOR RESPONSE

Response to Reviewer: 1, Dr. Pradeep Narayan, Bristol Heart Institute

Comments to the Author:

Overall, the authors have addressed the concerns raised and provided reasonable explanations. They also reference relevant literature to support their decisions and acknowledged the limitations.

There are few minor points:

1. Lines 336-338- the authors mention that –

In the APACHE cohort, female sex was the only variable that showed a significant correlation with COL1 α 1 concentrations in bivariate correlation analyses, though only at 6 months ($r = 0.19$, $p = 0.047$).

To be honest, the p-value if limited to 2 decimal points would become 0.05 and a $r = 0.19$ shows a very weak correlation. So, it might be best to rephrase the statement as it does not really show a “significant correlation”. Also, even though it is a very minor issue the p-value in the figure 2 is 0.048 but in the text it is 0.047.

Response: We agree on the comment that the correlation between female sex and COL1 α 1 in the APACHE cohort is weak in regard to statistical measures. However, the median difference between female and male sex in COL1 α 1 concentrations is approximately 20 percent. Furthermore, the median difference in COL1 α 1 concentrations between female and male sex in the LSH cohort were of the same magnitude (33 percent). Taken together, these findings indicate a potentially important difference between sexes, that has not previously been reported, and thus warrants further investigation. We have now rephrased the description of our findings in the Results section to emphasize that the correlation is weak.

We agree that a p-value of 0.047 can be rounded to 0.05 (2 decimal point). However, as stated in the JMIR Reporting Guidelines (please see reference below), the use of 3 decimal point is also acceptable as follows: “If $P > .01$ then the P value should always be expressed to 2 digits whether or not it is significant. When rounding, 3 digits is acceptable if rounding would change the significance of a value (eg, you may write $P = .049$ instead of $.05$).”

Reference: <https://support.jmir.org/hc/en-us/articles/360000002012-How-should-P-values-be-reported->

The p-values of 0.048 and 0.047 originate from two different statistical analyses, the Mann-Whitney U-test and the Spearman Correlation analysis, respectively. We have repeated these analyses and find the p-values of 0.0476 rounded to 0.048 and 0.0471 rounded to 0.047, consistent with the p-values reported in the manuscript.

2. The authors have defended the choice of the APACHE cohort by stating that they wanted to study this (COL1 α 1 biomarker both in a “pre-event” setting (LSH) and an “acute event” setting (APACHE).

In the conclusion, however, they only clearly report the findings of the LSH cohort. They should ideally summarize that COL1 α 1 is suited more as a pre-event marker rather than a marker that changes during an acute event.

Response: We agree on this comment. In the conclusion of the revised manuscript, we now state that COL1a1 seems to be of limited value in an acute coronary event setting.

3. Just out of interest it appears that all the biomarkers have been tabulated (table2) as median and IQR. Is it because the data was not normally distributed? Besides, it might be more informative to provide the data in the form Median (Quartile 1: quartile 3).

Response: Median and interquartile range (IQR) are used because of non-normal distributions of these variables. This is described in the statistical analysis subheading in the Methods section. We believe the use of IQR and quartile 1:quartile 3 are both acceptable.

VERSION 3 – REVIEW

REVIEWER	Narayan, Pradeep Bristol Heart Institute
REVIEW RETURNED	01-Sep-2023
GENERAL COMMENTS	Congratulations on a very nice study. The suggestion to use median with interquartile range was to provide a little more information than what the median and IQR provide. However, as the authors point out the use of IQR and quartile 1:quartile 3 are both acceptable. There is no need for any further changes to it.

VERSION 3 – AUTHOR RESPONSE